# Freezing Tolerance and Expression of β-amylase Gene in Two *Actinidia arguta* Cultivars with Seasonal Changes

**DOI:** 10.3390/plants9040515

**Published:** 2020-04-16

**Authors:** Shihang Sun, Jinbao Fang, Miaomiao Lin, Xiujuan Qi, Jinyong Chen, Ran Wang, Zhi Li, Yukuo Li, Abid Muhammad

**Affiliations:** 1Zhengzhou Fruit Research Institute, Chinese Academy of Agricultural Sciences, Zhengzhou 450009, China; 18838294775@163.com (S.S.); qixiujuan@caas.cn (X.Q.); chenjinyong@caas.cn (J.C.); wangran@caas.cn (R.W.); lizhi@caas.cn (Z.L.); liyukuo@caas.cn (Y.L.); Muhammadabid@caas.cn (A.M.); 2Key Laboratory of Horticultural Plant Biology (Ministry of Education), College of Horticulture and Forestry Science, Huazhong Agricultural University, Wuhan 430070, China

**Keywords:** kiwi fruit, freezing tolerance, electrolyte leakage, β-amylase gene

## Abstract

Low temperature causes injuries to plants during winter, thereby it affects kiwi fruit quality and yield. However, the changes in metabolites and gene expression during cold acclimation (CA) and deacclimation (DA) in kiwi fruit remain largely unknown. In this study, freezing tolerance, carbohydrate metabolism, and β-amylase gene expression in two *Actinidia arguta* cv. “CJ-1” and “RB-3” were detected from CA to DA stages. In all acclimation stages, the “CJ-1” was hardier than “RB-3” and possessed lower semi-lethal temperature (LT50). Furthermore, “CJ-1” had a more rapid acclimation speed than “RB-3”. Changes of starch, β-amylase, and soluble sugars were associated with freezing tolerance in both cultivars. Starch contents continued to follow a declining trend, while soluble sugars contents continuously accumulated in both cultivars during CA stages (from October to January). To investigate the possible molecular mechanism underlying cold response in *A. arguta*, in total, 16 *AcBAMs* genes for β-amylase were identified in the kiwi fruit genome. We carried out localization of chromosome, gene structure, the conserved motif, and the analysis of events in the duplication of genes from *AcBAMs*. Finally, a strong candidate gene named *AaBAM3* from *AcBAMs* was cloned in *Actinidia arguta (A. arguta),* The real-time qPCR showed that *AaBAM3* gene expression in seasonal changes was consistent with changes of soluble sugars. These results reveal that *AaBAM3* may enhance the freezing tolerance of *A. arguta* through increasing soluble sugar content.

## 1. Introduction

Low temperature is a major abiotic stress constraining plant growth, development, yield, and geographical distribution [1]. Plants have developed complicated mechanisms to adapt against stress with appropriate molecular, physiological, biochemical, and morphological changes [2]. Overwintering of perennial plants needs a dynamic development, including three development stages: acclimation (CA), freezing tolerance, and deacclimation (DA; loss of tolerance) [3]. For CA, plants always increase freezing tolerance after exposure to low non-freezing temperatures [4,5]. With regard to DA, plants will gradually lose freezing tolerance when placed under warm temperatures.

Plants can undergo many physiological and biochemical variations underlying seasonal changes [6]. These changes involve various pathways and ultimately increase freezing tolerance. The role of the starch metabolism pathway has been widely studied in plants. Starch, which is an important nutrient in plant growth and development, is not only used for essential life activities but is also involved in the response to abiotic stress. [7,8]. Soluble sugars play a major role in developing freezing tolerance, which has been investigated extensively in many plants. Plants accumulate soluble sugars when they are exposed to cold temperatures, which can lower osmotic potential, maintain turgor pressure in dehydrated cells, and allow plants to tolerate dehydrative stress [9]. Soluble sugars also act as cryoprotectants to safeguard the function of the plasma membranes and proteins under freezing stress. *Arabidopsis* plants showed sucrose accumulation during CA; a positive relation between sucrose and freezing tolerance has been observed in many plants [10]. A large number of studies have shown that plants with strong freezing tolerance had higher soluble sugars contents. The source of soluble sugars is generally originated from the metabolism of starch in plants [11]. The β-amylase is a major enzyme to hydrolyze starch. Moreover, β-amylase has a typical sugar-binding 14 domain (PF01373) at the N-terminus and is a member of glycosyl hydrolase family 14. Up till now, gene family analyses have identified 9 BAMs in *Arabidopsis*, 17 in pear, and 10 in rice [12]. Information on the expression of β-amylase genes in fruit plants is scarce, but in *Poncirus trifoliata*, some researchers indicated that the relative expression level of *BAM* genes increased significantly during CA, and activity of β-amylase had a positive correlation with freezing tolerance [13,14].

Kiwi fruit becomes an important fruit economically in recent years. It is mostly distributed in temperate and subtropical regions, which usually have low temperatures in the winter. The whole winter includes rapid temperature drops in earlier winter, freezing temperatures in midwinter, and abrupt low temperatures in late winter, which cause serious yield losses. Investigations about whole winter changes in freezing tolerance are important for the breeding of kiwi fruit cultivars with strong freezing tolerance.

*Actinidia* genus comprises of 54 species, and it is widely distributed from 0° to 50° north latitude in China [15]. *A. arguta*, which is distributed in the northeast of China, can tolerate low temperatures up to −40 °C. Currently, *Actinidia chinensis* and *Actinidia deliciosa* are the two main species that are commercially cultivated in the world. The natural distribution of these two species is to the south of the Yellow River, China (23–35° N). The freezing tolerance of these two species is weak and −13 °C in winter can cause serious damage to the plants [16]. The knowledge about the response of cold stress on the *Actinidia* is limited. Therefore, it is useful to reveal the mechanism of CA in *Actinidia* under cold stress.

In this study, we investigate carbohydrate metabolism and freezing tolerance under CA and DA stages of two cultivars for a better understanding of the mechanism of freezing tolerance in kiwi fruits. Additionally, we analyzed *AcBAMs* genes family and relative expression of the candidate gene to study the mechanism of freezing tolerance in depth. We hope this study will provide a theoretical basis for increasing the freezing tolerance of kiwi fruit.

## 2. Results

### 2.1. Seasonal Changes in Air Temperature

Winter temperatures in 2017 are shown in Figure 1. Temperature data for China came from an internet source (www.weather.com.cn). The temperature of sampling day was calculated by taking the daily mean temperature of 5 days. Air temperature from 23 October (S1) to 7 December (S3) fell sharply (Table 1). Between 22 December (S4) to 20 January (S6), there was the lowest temperature in the year, but the daily temperature represented fluctuations. Daily temperatures began to increase and gradually recovered to more than 0 °C after 20 January (S6). In general, the changes in winter temperature represented a U-shaped trend, which may be related to the variations in freezing tolerance of plants.

### 2.2. Freezing Tolerance in Two Actinidia Arguta

We can see that the changes in LT50 value corresponded with daily mean temperature, as shown in Figure 1. The values of LT50 varied significantly during winter from S1 to S7. For “RB-3”, the LT50 slowly declined over the period from S1 to S2. Due to the occurrence of sharper decline in LT50 at S2, it reached the lowest point at S5. The LT50 sharply rose again after S6. For “CJ-1”, there was a dramatic decline in the LT50 from S1 to S3. The LT50 changes showed a similar U-shaped pattern in both cultivars, but the curve for “CJ-1” was lower than the curve for “RB-3”. Although the “CJ-1” showed an earlier decline in LT50 than “RB-3” did, it showed a sharper rise than RB-3 after S6.

### 2.3. Metabolite Contents

The changes in the contents of starch over the period from S1 to S7 are given in Figure 2A,D, and we found a general downward trend. In the case of “RB-3”, the contents of starch decreased significantly from S1 to S3, whereas it slightly decreased from S3 to S6. Furthermore, it reached to the lowest point at S7. Additionally, the “CJ-1” and “RB-3” followed the same trend and details for it.

The cultivars showed dome-shaped trend for the activity of β-amylase (Figure 2B,E). For “RB-3”, β-amylase activity continued to rise steadily and reached a peak at S6. Afterwards, β-amylase activity showed steady decline. For “CJ-1”, we found higher similarities between “RB-3” and “CJ-1” for β-amylase activity during the seasonal changes. However, its activity was higher in “CJ-1” than “RB-3” over the seasonal changes.

Soluble sugars contents in shoots are given in Figure 2C,F. In “RB-3”, soluble sugars contents in both S1 and S2 periods remained low with no significant difference. Moreover, it had sharp rise from S2 to S4 and maintained a high level from S4 to S6. Furthermore, S7 had a sharp drop in soluble sugars contents and these contents reached to the lowest level. In the case of “CJ-1”, soluble sugars contents in the shoot tissue showed a gradual increase from S1 to S6 while they decreased at S7. In general, the soluble sugars of the two cultivars showed the dome-shaped trend during the change of winter temperature. However, “CJ-1” accumulated soluble sugars earlier than “RB-3”.

### 2.4. Bioinformatic Analysis of AcBAM

We used the *A. chinensis* genome instead of the *A. arguta* genome because it was unknown. All the 16 genes were identified for the *BAM* gene family in kiwi fruit. Distributions of *AcBAM* genes on 11 chromosomes of kiwi fruit were unequally demonstrated (Figure 3). Chromosome 25 was found to have the largest number of *AcBAM* genes (3 genes), while chromosomes 8, 18, 24 had two *AcBAM* genes, respectively, and the remaining chromosome only had one *AcBAM* gene. Moreover, the length of the specified protein of *AcBAM* ranged from 317–697 amino acids and their molecular weights (MW) ranged from 35.74 to 77.79 kDa, and their theoretical iso-electric point (PI) ranged from 5.51 to 8.72 (Table 2).

The exon/intron structure analysis was of great significance in phylogenetic studies of gene families. The members of the *AcBAM* family demonstrated similar exon/intron structures based on gene structure analysis (Figure 4). Generally, the number of exons range from 1 to 10. In this study, the exon/intron structure of 16 *AcBAM* genes was systematically analyzed, and it was found that all *AcBAM* genes had abundant introns. Moreover, the exon/intron structure of each gene was highly conserved. In addition, intron gain/loss was also found in the structural analysis. For example, the PSS01267.1 gene is composed of 9 exons, while PSR93503.1 is composed of 3 exons. The conserved Glycosyl hydrolase family domain 14 (PF01373) was identified by Pfam for BAM proteins. Additionally, we identified 10 distinct motifs by MEME. The results showed that 16 identified *AcBAMs* in this study had conserved motif regions.

Analysis of synteny and gene duplication of *AcBAMs* in kiwi fruit for the tandem and segmental duplication of *AcBAM* genes were surveyed through 11 chromosomes of kiwi fruit (Figure 5). Among the 16 *AcBAM* genes, there were genome-wide replication events. The *AcBAM* gene on chromosome 14 had a replication relationship with the *AcBAM* on chromosome 24, indicating that the *AcBAM* genes in the genome had conservative characteristics. In addition, two chromosomes on chromosomes 18 and 24 have tandem repeats.

To assess the evolutionary associations of *AcBAM* genes, phylogenetic analysis was performed by aligning 16, 9, 1, 1, 1, and 1 BAM proteins from kiwi fruit, *Arabidopsis*, pear, tea, blueberry, and orange. The maximum likelihood tree categorized the studied genes into four main clades, A–D. We found that clades A belonged to AtBAM3 type, which contained five AcBAMs, one PbrBAM3, one PtrBAM1, one VcBAM3, and one CsBAM3. Clades B belonged to the AtBAM1 type, which contained three AcBAMs. Clades C contained two AcBAMs, and Clades D contained six (Figure 6).

The five AcBAMs from clades A include PSS11600.1, PSR93503.1, PSR93504.1, PSS09671.1, an PSR91672.1. In addition, they share more than 90% sequence similarity (Figure 7). According to 5 *AcBAM* genes sequences, we designed primers to clone the full-length *AaBAM* gene in *A. arguta*. Due to the genomic differences between the *Actinidia arguta* and *Actinidia chinensis*, we obtained only one *BAM* gene by homologous cloning. Additionally, this one *BAM* gene was named *AaBAM3*, and it had a full-length of 1644 bp (Genebank Accession Num. 2327905).

### 2.5. Expression Analysis of BAM

The β-amylase is a key enzyme in the glucose metabolism pathway that can degrade starch into maltose to increase the soluble sugars contents in plants. Hence, we designed fluorescence quantitative primers on the bases of the specificity of this gene (Table 3). The expressions of the two cultivars were performed in 7 stages of variation in temperature over time (S1–S7). The highest expression level of the *AaBAM3* gene appeared in “RB-3” at S6 that followed a dome-shaped trend. Similarly, “CJ-1” followed the same dome-shaped trend for relative expression of the *AaBAM3* gene. Furthermore, the rise in relative expression was more prominent between S3 and S6, and it declined significantly at S7 (Figure 8).

## 3. Discussion

The plants have developed the capacity to tolerate subfreezing temperatures in order to survive under cold stress. Generally, cold acclimation in plants is triggered by short days and low temperatures. In the beginning, short days cause partial acclimation, and low temperatures lead to a further increase in freezing tolerance. Finally, warm temperatures lead to a loss in freezing tolerance or deacclimation [17]. Our seasonal experiment started in October when both cultivars had the same cold tolerance level (around 0 °C). After acclimation, the LT50 evaluation results indicated that “CJ-1” was more freezing tolerant than “RB-3”. We found our results consistent with filed observations. Moreover, we observed different cold acclimation patterns for cultivars: freezing tolerance was established early in “CJ-1”, while cold acclimation in “RB-3” started later, which may lead the plants to cold injury during early winter. The cultivar with high freezing tolerance acquired partial cold hardiness before the onset of the lowest temperatures. The cultivar “CJ-1” showed more rapid deacclimation than “RB-3” at S6, whereas “RB-3” was hardier than “CJ-1” at the S7 stage, which was quite the opposite to our expectation. There might be a relationship between deacclimaton and chilling requirements. However, from field observations, we found that the hardier cultivar under mid-winter always suffers from injuries during the low-temperature exposure of spring. The warm temperatures of spring result in the loss of cold hardiness, and deacclimation occurs faster than acclimation. Hence, we speculated that the hardier cultivar might be more sensitive to temperature changes. The hardier cultivar may have reached its dormant stage before the exposure of low temperature, and then immediately broke the dormancy in spring to resume early growth with the rise in warmer temperatures.

Freezing tolerance in shoots of the two kiwi fruit cultivars was rapidly increased during CA and decreased during DA as in other woody perennials [18]. Although the patterns of freezing tolerance were different in both cultivars, yet the “CJ-1” was hardier than “RB-3” during CA, which may be related to a genetic mechanism. In our past unpublished data, the 20 *A. arguta* genotypes exhibited a wide range of freezing tolerances between −18.20 to −35.23 °C based on LT50 values. Gererally, the two *A. arguta* genotypes had the same genetic background, but different transcriptional regulatory pathways can cause different freezing tolerances in the same species.

The starch-to-sugar conversion is an important parameter for studying freezing stress in perennial plants [19,20]. Changes in soluble sugars content are generally accompanied by changes in starch content. Soluble sugars acted as osmoticum that reduces cellular dehydration. Moreover, soluble sugars serve as a carbon source for cellular metabolism to directly protect macromolecules [21]. For both cultivars, an increase of soluble sugars correlated with LT50 levels. The decrease in starch content from S1 to S6 was associated with the increases of soluble sugars content via the hydrolysis of starch into soluble sugars that helped in enhancing freezing tolerance. From S6 to S7 stage, the activity of β-amylase and starch contents showed a large decrease. These phenomena implied that β-amylase was not involved in growth resumption. In spite of the decrease of starch content after S7, the soluble sugars contents were also decreased. These observations might be due to the changes in metabolic processes such as respiration of deciduous woody perennial for preparing themself for growth resumption. The greater activity of β-amylase in “CJ-1” than in “RB-3” coincided with higher *AaBAM3* expression in “CJ-1” than in “RB-3”, which represented higher starch metabolic activities in “CJ-1”. The higher starch metabolic activities appear to play an important role in resisting freezing injury.

This study, through whole-genome bioinformatics analysis, predicted 16 members of the family of *AcBAM* genes. The gene family contained a conserved glycosyl hydrolase family 14 domain (PF01373). The number of genes and the classification were similar to that of *Arabidopsis* and rice. The *BAM* gene sequences, both in monocots and dicotyledonous plants, were relatively conserved [22]. A gene family was generated by the duplication and variation of ancestral genes. It is of great significance to study the systematic evolution of gene families and analyze the gene structure and composition of family genes. The results showed that all members of the *AcBAM* gene family were composed of exons and introns. The number of introns in *Arabidopsis thaliana* family ranged from 2 to 9, and the number of introns in rice ranged from 1 to 10 [23,24]. Evolutionary analysis showed that in the four subfamilies, the members of the family with the closest evolutionary relationship clustered together, and their genetic structure and amino acid composition were similar. In detail, we analyze gene evolution, genetic structure, and conservative motif in kiwi fruit *BAM* gene family.

*BAM* gene is involved in the hydrolysis of starch under low-temperature stress [25], and the RT-qPCR results of this study showed that the expression of the *AaBAM3* gene was positively correlated with the activity of β-amylase. The two different soluble sugars and starch contents in the two kiwifruit cultivars were affected by the differential expression of *AaBAM3*. The relative expression of *AaBAM3* initially increased, then decreased, which was similar to the change in β-amylase, and this justifies the change in relative expression levels in *AaBAM3*. Moreover, the relative expression of *AaBAM3* was upregulated threefold in “CJ-1” than in “RB-3” during the S6 stage, which suggested that *AaBAM3* was the major gene responsible for the different freezing tolerance of the cultivars. Different *BAM3* gene expressions in different cultivars caused different freezing tolerance. The same results have previously been reported in blueberry. The overall results of starch contents, β-amylase activities, soluble sugars, and *AaBAM3* relative expression levels suggest that *AaBAM3* is directly involved in the degradation of starch to soluble sugars that are important for enhancing plant cold-hardiness.

Based on our results, we showed a cold response model for kiwi fruit (Figure 9). The kiwi fruit stayed dormant from S1 to S6 stages. Initially, plant cells receive the cold signal, then *AaBAM3* is upregulated to increase the activity of β-amylase, and it hydrolyzes starch to yield soluble sugars. Finally, soluble sugars act as osmoticum and improve the freezing tolerance of kiwi fruit by reducing cellular dehydration. The kiwi fruit starts growth resumption from S6 to S7 stages, and *AaBAM3* is downregulated to decrease the activity of β-amylase that stops the starch-β amylase-soluble metabolism pathway. Finally, kiwi fruit loses freezing tolerance.

## 4. Materials and Methods

### 4.1. Plant Material and Experimental Conditions

Two *A. arguta* cultivars (“CJ-1” and “RB-3”) were selected as materials based on a preliminary freezing tolerance screening test [26]. The “CJ-1” came from Liaoning (126°35′ E 46°40′ N) province; “RB-3” came from Henan (111°27′ E 33°81′ N) province. The annual extreme temperature was around −26 °C (Liaoning) and −5 °C (Henan) in each province. Both two cultivars “CJ-1” and “RB-3” were obtained from Zhengzhou Fruit Research Institute Kiwifruit Germplasm Repository.

Shoots were sampled during mid-October 2017 until March 2018. The detailed data can be seen in Table 1. After sampling, samples were put into an ice-box on the way to the laboratory. Then, controlled freezing tests and shoot physiological indices were determined using fresh samples. The remaining samples were frozen at −80 °C for gene expression analysis. 

### 4.2. Determinations of Freezing Tolerance

The detached shoots of kiwi fruit were exposed to low temperatures according to the previous methods [27]. The detached shoots were rinsed with double-distilled water to remove surface contaminants. The middle section of the shoots, about 10–15 cm long, were wrapped with polyethylene film and placed in a programmable freezing chamber. The temperature was decreased stepwise. The materials were exposed to temperatures of −5, −10, −15, −20, −25, and −30 °C for 8 h each. After low temperature treatment, the shoots without buds were cut into 1–2 mm thick slices. Approximate 0.2 g of the slices were incubated in 30 mL of double-distilled water for 2 h, with shaking at 200 rpm at room temperature. The initial electrical conductivity (*C1*) was measured using a digital conductivity meter (DDS-307, Rex, Shanghai, China). The samples were heated up to boiling for 30 min, then cooled down to room temperature with shaking for 30 min. The second electrical conductivity (*C2*) was then taken. The *REL* was calculated as indicated by Equation (1):*REL* (%) = (*C*_1_/*C*_2_) × 100%(1)

The freezing tolerance was expressed as LT50 (lethal temperature at which *REL* reaches 50%) by fitting the response curve obtained by the *REL* with a logistic sigmoid function (Equation (2)):*Y* = *k*/(1 + *ae*^−*bx*^)(2)
where *x* is the treatment temperature, *y* is the *REL* value, *k* indicates the extreme value when *x* approaches infinity, *a* and *b* are the equation parameters. If the correlation coefficient r is close to 1, the equation is used to calculate LT50 [28].

### 4.3. The Measurement of Carbohydrate Metabolites Contents

The shoots were ground using an electric grinding mill. Approximate 0.2 g of the sample powder was placed into a 2-mL tube; the frozen powder was immediately used for extraction. Starch content was determined using the acid hydrolysis test [29]. The activity of β-amylase was determined using the DNS (3,5-dinitrosalicylic acid) method. The content of soluble sugars was determined using the anthrone colorimetry [30]. The measurements were repeated 3 times.

### 4.4. RNA Extraction and Reverse Transcription

Total RNA was isolated from shoots (100 mg) using a Plant Total RNA purification kit (Waryoung, Beijing, China) following the manufacture’s instructions. RNA quality and concentration were detected with a NanoDrop spectrophotometer. RNA samples were converted to cDNA using a single-stranded cDNA synthesis kit (TOYOBO, Osaka, Japan).

### 4.5. Analysis of AcBAM Gene Family

The *BAM* gene family was identified in the kiwi fruit genome. The sequences were downloaded from *Actinidia chinensis* “Red5” genome [31]. The downloaded file was used for downstream analysis. The Glyco_hydro_14 domain (PF01373) obtained from the PFAM database was utilized as a query for the hidden Markov model (HMM) looking, utilizing the HMMER3.0 program with a pre-characterized limit of E < 1e^−5^. Moreover, the conservation domain of the Glyco_hydro_14 domain was utilized as a query to search against the “Red5” protein dataset using the BLASTP program with an estimation of 1e^−5^ as the threshold. Comparing the HMM results and BLASTP results, after a manual amendment and removing the mistake results, the remaining sequences were the putative *AcBAM* proteins. Moreover, we use the PFAM web server to further confirm the putative *AcBAM* genes. Then, the analysis of the number of amino acids, molecular weight, and pI in the kiwi fruit BAM was conducted.

Phylogenetic analysis. MEGA ver.10.0.5 was utilized to perform multiple sequence alignment and maximum likelihood tree with 1000 bootstrap replications.

The chromosome distributions of *AcBAM* genes were acquired from the genome annotation information. Conserved motifs and domains were analyzed using the MEME Suite web server. The physical distribution of *AcBAM* genes on chromosomes was illustrated by TBtools based on gene position in the genome.

For syntenic analysis, duplications between the *Actinidia chinesis BAM* gene among kiwi fruit were obtain from MCScan, and diagrams were drawn by TBtools.

### 4.6. Isolation and Expression of Genes

Candidate *AcBAM* genes were selected from previous bioinformatic analysis. Using Primer Premier 5, primers were designed. RT-qPCR analysis was conducted, as described in Yukuo Li [32] with minor modification. Candidate reference genes *AcBAM* were analyzed. SYBR Green-based RT-qPCR analyses were performed in a LightCycler 480 (Roche480) on a 96-well plate. The programs for the PCR amplifications were following: 95 °C for 5 min, followed by 45 cycles of 10 s at 95 °C, 20 s at 60 °C, and 20 s at 72 °C. A melt-curve analysis was carried out using the default parameters (5 s, 95 °C and 1 min, 65 °C). We used the β-actin in the kiwi fruit as the control gene to normalize. All analyses had three biological replicates. The relative expressions were calculated using the 2^−ΔΔCt^ method. RT-qPCR primers of target genes were designed using Primer Premier 5 software.

### 4.7. Statistical Analysis

Physiological data (LT50, metabolite contents) were subjected to one-way ANOVA. Statistical analyses were performed in SPSS software (ver. 22).

## 5. Conclusions

Freezing tolerance of two *A. arguta* cultivars was associated with the starch-soluble sugars metabolic pathway, which partly attributed to the expression of the *AaBAM3 gene*. The two cultivars from *A. arguta* had the same freezing tolerance before cold acclimation, and both were sensitive to freezing stress. However, the freezing tolerance increased in both cultivars in mid-winter and “CJ-1” became more freeze-tolerant than “RB-3”. Additionally, “CJ-1” acclimated faster than “RB-3”.

Furthermore, we obtained 16 *AcBAM* genes after conducting an extensive genome-wide search and compared their exon–intron structure, the motif combination, and the phylogenetic relationship In addition, the *AcBAMs* were clustered into four clades with regard to their enzyme activity. We suggest the *AaBAM3* gene as a candidate gene for further study on cold stress. 

## Figures and Tables

**Figure 1 plants-09-00515-f001:**
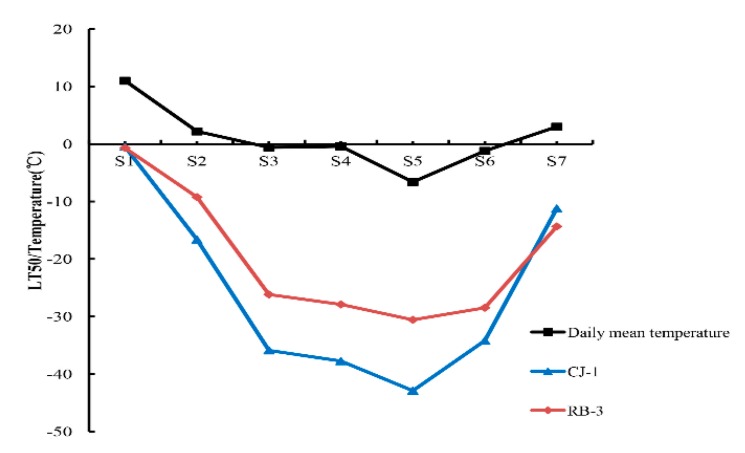
Seasonal changes in air temperatures data at the sampling site (111°27′E, 33°81′N) from October 2017 to March 2018 and freezing tolerance estimated as the half lethal temperature (LT50) in the shoots of “CJ-1” and “RB-3” kiwifruit cultivars.

**Figure 2 plants-09-00515-f002:**
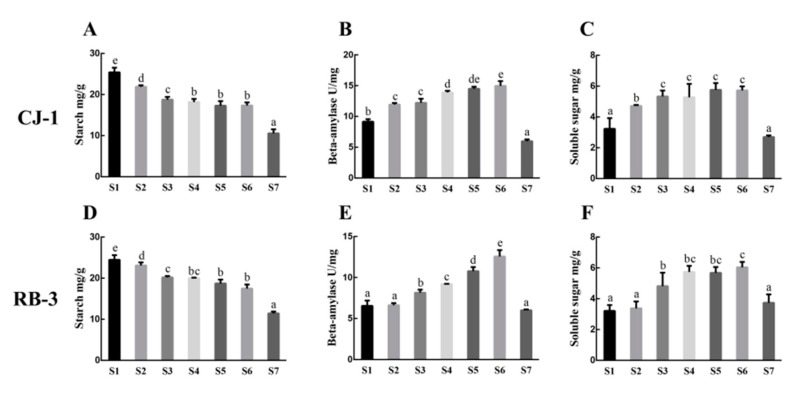
Seasonal changes in starch contents (**A**,**D**), β-amylase activity (**B**,**E**), and soluble sugars contents (**C**,**F**) for shoots of “CJ-1” and “RB-3”. Data are present as means ± standard errors (*n* = 3).

**Figure 3 plants-09-00515-f003:**
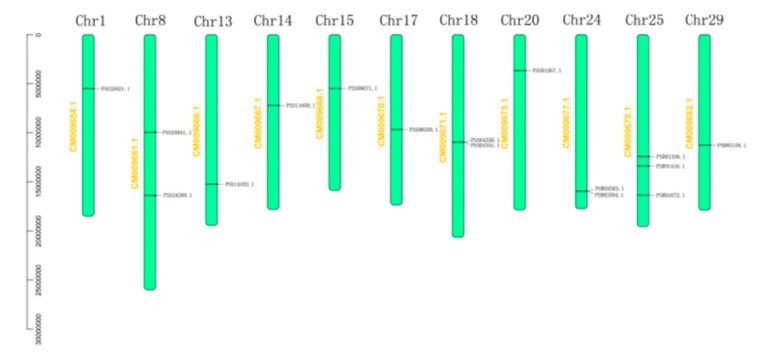
Localization of the *BAM* genes in *A. chinensis* genome. Chromosome number is dictated on the chromosome.

**Figure 4 plants-09-00515-f004:**
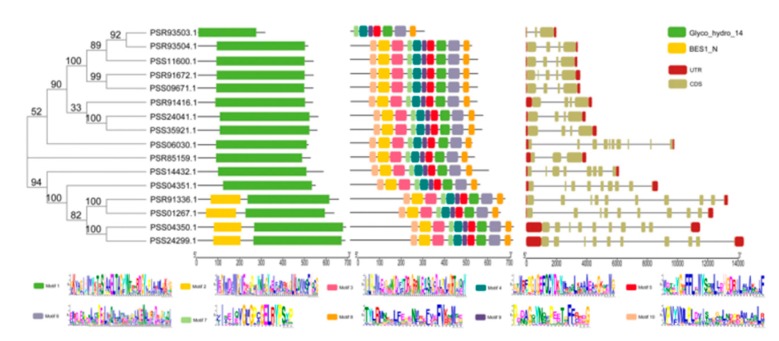
Consensus phylogenetic tree of *BAM* genes in *A. chinensis* constructed by amino acid multiple sequence alignment using MEGA-X software. Schematic representations of the conservative domain of *AcBAM* genes. Schematic representations of conservative motifs of *AcBAM* genes. Different motifs are highlighted with different colored boxes with numbers 1 to 20. Lines represent protein regions without detected motifs. Exons are represented by boxes. Dashed line connecting two exons represents an intron.

**Figure 5 plants-09-00515-f005:**
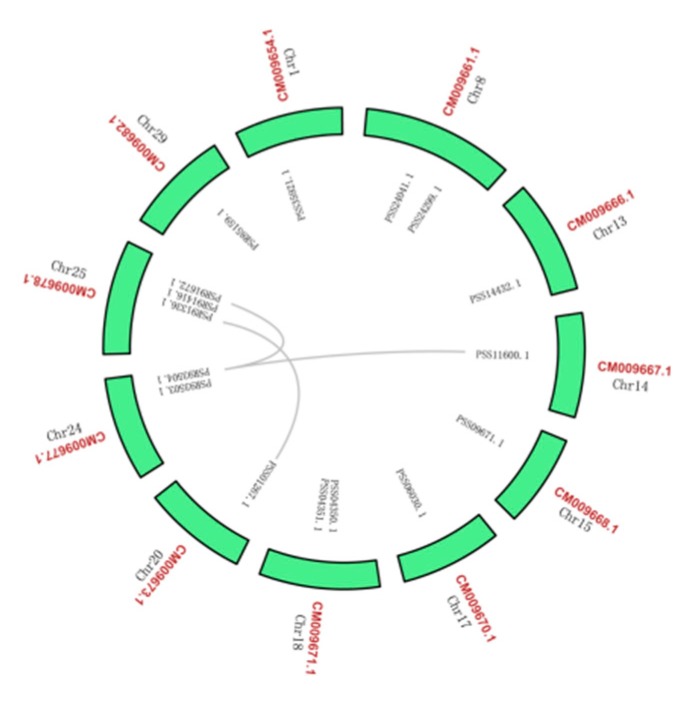
Duplication of the *BAM* genes in the *A. chinensis* genome. The synteny relationship between each pair of *BAM* genes was detected using linear regression. Genes with a asyntenic relationship are linked by grey lines.

**Figure 6 plants-09-00515-f006:**
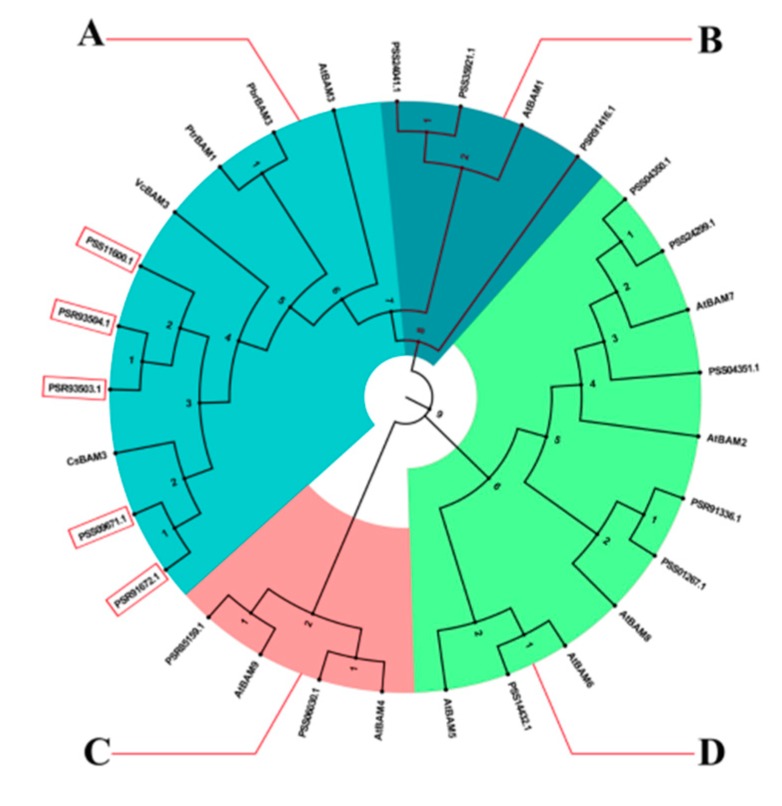
Phylogenetic trees of BAM genes in *A. chinensis*, *Camellia sinensis*, *Citrus trifoliata*, *Vaccinium corymbosum*, *Pyrus bretschneideri,* and *A. thaliana* based on maximum likelihood analysis of the BAM domain amino acid sequence alignment.

**Figure 7 plants-09-00515-f007:**
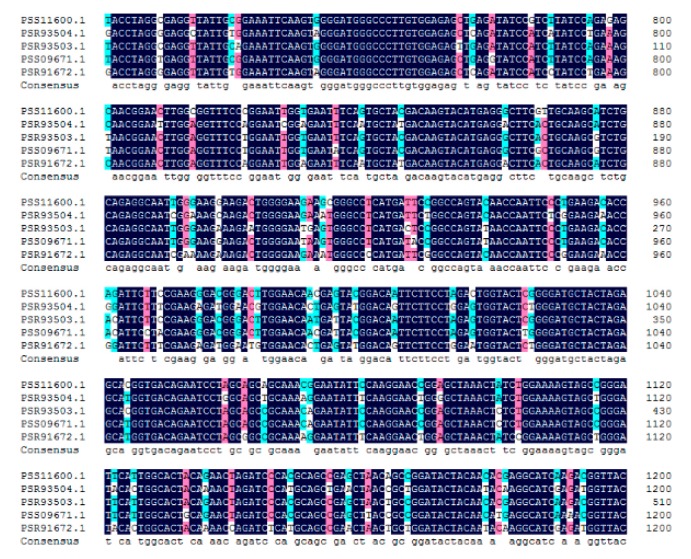
Alignment of the core sequence of part *AcBAMs* and their consensus sequences.

**Figure 8 plants-09-00515-f008:**
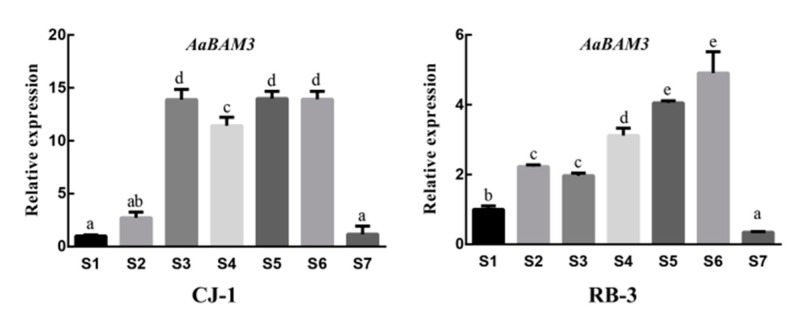
Seasonal changes in the relative gene expression of *AaBAM3* in the shoots of “CJ-1” and “RB-3” kiwifruit cultivars. All values were normalized to the expression levels of β-ACTIN housekeeping gene. Data were presented as means ± standard errors (*n* = 3).

**Figure 9 plants-09-00515-f009:**
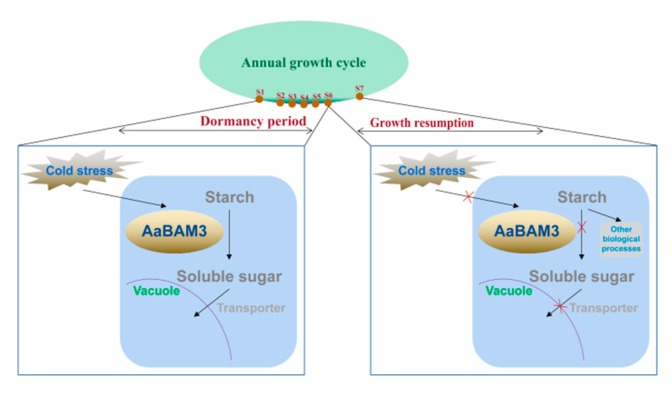
The proposed model of *AaBAM3*-mediated cold tolerance in kiwi fruit. The cold induces the expression of *AaBAM3* and β-amylase hydrolyzes starch to yield soluble sugars, thereby improving plant freezing tolerance.

**Table 1 plants-09-00515-t001:** Sampling time corresponding to phase.

Date	23 October	22 November	7 December	22 December	5 January	20 January	5 March
Phase	S1	S2	S3	S4	S5	S6	S7

**Table 2 plants-09-00515-t002:** The basic information of the *AcBAM* gene family.

Accession	Location	pI	Molecular Weight (Da)	Size (aa)
PSR85159.1	Chr29: 11230497–11233893	6.71	58895.73	532
PSR91336.1	Chr25: 12382772–12394190	5.76	74551.84	664
PSR91416.1	Chr25: 13363180–13366908	6.8	61047.39	542
PSR91672.1	Chr25: 16324193–16327258	7.22	61707.94	546
PSR93503.1	Chr24: 15922030–15923754	8.52	35748.08	317
PSR93504.1	Chr24: 15932333–15935251	8.72	59006.06	520
PSS01267.1	Chr20: 3639423–3650017	6.02	72661.93	644
PSS04350.1	Chr18: 10923906–10933753	5.24	78372.16	700
PSS04351.1	Chr18: 10935330–10942790	5.46	63106.07	556
PSS06030.1	Chr17: 9640160–9648546	8.83	59864.53	523
PSS09671.1	Chr15: 5454601–5457662	7.56	61368.5	546
PSS11600.1	Chr14: 7186173–7189084	8.76	61812.1	547
PSS14432.1	Chr13: 15225015–15230273	5.51	66860.73	594
PSS24041.1	Chr8: 9912429–9915793	5.78	63364.85	569
PSS24299.1	Chr8:16339786–16352100	5.6	77792.48	697
PSS35921.1	Chr1: 5464816–5468823	6.06	62737.09	565

**Table 3 plants-09-00515-t003:** The sequences of primers used in this paper.

Primer Name	Sequence	Type
AaBAM3-F	5′-ATGAATGCCAGCTTGATGGC-3′	PCR
AaBAM3-R	5′-TTACACTAGAGCAGCCTCCTTCG-3′	
qBAM-F	5′-ACCTACCAATAGCGCGGATG-3′	RT-qPCR
qBAM-R	5′-AACTAACCCTTCTGGCGAGC-3′	

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
