# Peer review of "Freezing Tolerance and Expression of β-amylase Gene in Two *Actinidia arguta* Cultivars with Seasonal Changes"

_plants, 2020, doi:10.3390/plants9040515_

Round 1
Reviewer 1 Report
The presented experimental work is of high quality. The methodology is understandable, the interpretation of the results is at a very high level, clearly demonstrating the drawn conclusions. Graphic attachments (tables, graphs, and especially pictures) are above-standard and support credibly the found results. The discussion logically comments the results and compares them adequately with the published data. Although the work is quite extensive, it is very clear and easy to read.
I have no serious comments on the submitted paper.
I would just recommend if possible:
In the following text, it is necessary to modify the Latin plant names to italics:
212 Figure 6. phylogenetic trees of BAM genes in A. chinensis, Camellia sinensis, Citrus trifoliata, Vaccinium corymbosum,
213 Pyrus bretschneideri and A. thaliana based on maximum likelihood analysis of the BAM domain amino acid sequence
214 alignment.
Author Response
Dear Reviewer:
Thank you for taking the time to review our manuscript entitled ‘Freezing Tolerance and Expression of β-amylase Gene in Two Actinidia arguta Cultivars with Seasonal Changes’ ( ID: plants-773932) and give the comments. These comments are all valuable and very helpful for revising and improving our paper. We have studied comments very carefully and made correction, which we hope meet with approval. The main corrections in the paper and the response to the comments are as following:
Comment: It is necessary to modify the Latin plant names to italics: Figure 6. phylogenetic trees of BAM genes in A. chinensis, Camellia sinensis, Citrus trifoliata, Vaccinium corymbosum, Pyrus bretschneideri and A. thaliana based on maximum likelihood analysis of the BAM domain amino acid sequence alignment.
Response: We are very sorry for our incorrect writing. According to your instruction, we have changed ‘Figure 6. phylogenetic trees of BAM genes in A. chinensis, Camellia sinensis, Citrus trifoliata, Vaccinium corymbosum, Pyrus bretschneideri and A. thaliana based on maximum likelihood analysis of the BAM domain amino acid sequence alignment.’.
We tried our best to improve the manuscript and made some changes in the manuscript. We appreciate your warm earnestly, and hope that the correction will meet with approval.
Once again, thank you very much for your comments and suggestions.
Reviewer 2 Report
Nice work!
I suggest some minor revisions regarding introduction and bacgroud section. Also, there are some adjustments needed regarding the Table 3 placement...
Author Response
Dear Review:
Thank you for taking the time to review our manuscript entitled ‘Freezing Tolerance and Expression of β-amylase Gene in Two Actinidia arguta Cultivars with Seasonal Changes’ ( ID: plants-773932) and giving the comments. These comments are all valuable and very helpful for revising and improving our paper. We have studied comments very carefully and have made correction which we hope meet with approval. The main corrections in the paper and the response to the comments are as following:
Comment 1: I suggest some minor revisions regarding introduction and backgroud section.
Response 1: We are very sorry for our imperfect pressentation of instroduction. Carefully considering your suggestions, we do some minor revisions as following:
Introduction
Line 30: ‘environmental’ changed to ‘abiotic’. Delete ‘crop’.
Line 31: ‘elaborate’ changed to ‘complicated’; ‘to’ changed to ‘against’.
Line 33: ‘plant is a….main stages:’ changed to ‘plants need a dynamic development, including three development stages’.
Line 34: ‘following exposure’ changed to ‘after exposing’.
Line 35-36: ‘when plants …gradually lose’ changed to ‘plants will gradually lose freezing tolerance when placed under warm temperatures.’.
Line 37: ‘Seasonal changes in …’ changed to ‘Plants can undergo many physiological and biochemical variations underlying seasonal changes’.
Line 39-40: ‘has been widely…’ changed to ‘studied in plants. Starch that is an important nutrient in plant growth and development is not only used for essential life activities, but also involves in responsing to abiotic stress.’ .
Line 41: ‘sugar’ changed to ‘sugars’.
Line 42-44: ‘extensively in many…’ changed to ‘Plants accumulate soluble sugars when they are exposed to cold temperatures, which can lower osmotic potential, maintain turgor pressure in dehydrated cells and allow plants to tolerate dehydrative stress’.
Line 44: ‘sugar’ changed to ‘sugars’.
Line 44-45: ‘protect the plasma membranes…’ changed to ‘safeguard function of the plasma membranes and proteins’.
Line 45: ‘…CA in Arabidopsis’ changed to ‘Arabidopsis plants showed sucrose accumulation during CA, a positive relation’.
Line 47: ‘tolerance have’ changed to ‘tolerance had’; ‘sugar content’ changed to ‘sugars contents’.
Line 48: ‘sugar’ changed ‘sugars’; ‘β-amylase’ changed to ‘The β-amylase’.
Line 49-50: ‘Moreover, β-amylase ...9 BAMs’ changed to ‘Moreover, β-amylase has a typical sugar-binding 14 domain (PF01373) at the N-terminus, and is a member of glycosyl hydrolase family 14. Up till now, gene family analyses have identified 9 BAMs’.
Line 52: ‘gene’ changed to ‘genes’; ‘limited’ changed to ‘scarce’.
Line 53: ‘β-amylase’ changed to ‘BAM genes’;
Line 54: ‘and was correlated with freezing tolerance‘ changed to ‘and activity of β-amylase had positive correlation with freezing tolerance’.
Line 55- 59: Add kiwifruit background.
Line 60: ‘comprises’ changed to ‘comprises on’.
Line 61: ‘A. arguta,…’ changed to ‘A. arguta which is‘; ‘down’ changed to ‘up’.
Line 63: ‘in’ changed to ‘on’.
Line 64: ‘could’ changed to ‘can’.
Line 65: ‘The knowledge on response of the Actinidia to cold stress’ changed to ‘The knowledge about response of cold stress on the Actinidia’.
Line 66: ‘the cold acclimation mechanism of’ changed to ‘mechanism of CA in’.
Line 67-71: whole paragraph changed to ‘In this study, we investigated carbohydrate metabolism and freezing tolerance under CA and DA stages of two cultivars for a better understanding of mechanism of freezing tolerance in kiwifruits.. Additionally, we analyzed AcBAMs genes family and relative expression of candidate gene to study mechanism of freezing tolerance in depth. We hope this study will provide theoretical basis for increasing freezing tolerance of kiwifruit.’.
Comment 2: There are some adjustments needed regarding the Table 3 placement.
Response 2: Thank you very much for your comment on the Table 3 placement, Table 3 has been put into line 282.
We tried our best to improve the manuscript and made some changes in the manuscript. We appreciate your warm earnestly, and hope that the correction will meet with approval.
Once again, thank you very much for your comments and suggestions.
Reviewer 3 Report
See attached file.

Author Response
Dear Reviewer:
Thank you for taking the time to review our manuscript entitled ‘Freezing Tolerance and Expression of β-amylase Gene in Two Actinidia arguta Cultivars with Seasonal Changes’ ( ID: plants-773932) and giving the comments. These comments are all valuable and very helpful for revising and improving our paper. We have studied comments very carefully and have made correction which we hope meet with approval. The main corrections in the paper and the response to the comments are as following:
Major comment 1: The authors should be careful to say that they have only found a correlation between these levels and have not proven a cause and effect relationship. Expression of many genes and metabolites change in response to low temperature treatment, but which ones are the most important is still really unknown.
Response 1: It is exactly as you said, freezing tolerance is a quantitative trait. Many pathways can contribute to increase freezing tolerance. The well-known CBF regulon have been outlined, however, at least 28 % of cold-responsive genes were not affected by expression of the CBF transcription factors. In our study, we only found a correlation between metabolites and gene, which was deficient to directly lead to increase freezing tolerance. We think molecular-biology experiments (such as: gene over-expression or gene silence) are necessary to clarify this question. In future, we will carry out many experiments to solve this question.
Major comment 2: What is the chilling requirements of these cultivars? If known, the authors should discuss this in relation to the deacclimation data. The hardier cultivar could have a shorter chilling requirement and, thus, respond faster to warming temperatures.
Response 2: We are very sorry, we did not know their chilling requirements about ‘CJ-1’ and ‘RB-3’. However, we found a paper that described chilling requirements of different Actinidia arguta through reading references. Results showed there were divergence in different Actinidia arguta cultivars, some genotypes had 672 h in chilling requirements, some genotypes had 1056 h in chilling requirements. We think you are right, deacclimaton have a correlation with chilling requirement. Next step, we will focus on chilling requirements about ‘CJ-1’ and ‘RB-3’.
Major comment 3: The manuscript needs to be edited extensively for grammar, preferably by an English speaking colleague.
Response 3: We are very sorry for our imperfect writing. According to your detail instruction, we have changed one by one. Based on our too many incorrections, we invite our colleague native English speaker to revise our manuscript. We do our best to improve our grammar. The detail corrections are as following:
Minor comment :
Title
Line 3: ‘in’ changed to ‘with’.
Line 4: deleted ‘and’.
Line 5: added ‘and Abid Muhammad1’.
Abstract
Line 11: ‘plant’ changed to ‘plants’; ‘thereby affecting’ changed to ‘thereby it affects’.
Line 15 -16: ‘‘CJ-1’ was hardier than ‘RB-3’, ‘CJ-1’ has lower semi-lethal temperature (LT50) than ‘RB-3’ in all acclimation stages’ changed to ‘In all acclimation stages, the ‘CJ-1’ was hardier than ‘RB-3’ and possessed lower semi-lethal temperature (LT50).’.
Line 16: ‘especially, ‘CJ-1’ has more’ changed to ‘Furthermore, ‘CJ-1’ had a more’.
Line 17: ‘soluble sugar’ changed to ‘soluble sugars’; ‘are’ changed to ‘were’.
Line 18: ‘contents are keeping decline trend. Soluble sugar’ changed to ‘contents continued to follow declining trend, while soluble sugars’; ‘accumulate’ changed to ‘accumulated’.
Line 18-19: ‘in the two types’ changed to ‘in both cultivars’.
Line 20: ‘β-amylase genes are identified in kiwifruit genome, a total of 16 AcBAMs were obtained’ changed to ‘in total 16 AcBAMs gene for β-amylase were identified in kiwifruit genome.’.
Line 21: ‘Location of…’ changed to ‘We carried out localization of…’.
Line 22: ‘for’ changed to ‘from’; ‘a strongly candidate BAM gene is cloned in Actinidia arguta (A. arguta), named AaBAM3’ changed to ‘a strong candidate gene named AaBAM3 from AcBAM was cloned in Actinidia arguta (A. arguta)’.
Line 23: ‘Real-time qPCR showed AaBAM3’ changed to ‘The Real-time qPCR showed that AaBAM3 ’
Line 24: ‘are’ changed to ‘was’; ‘This’ change to ‘These’.
Line 25: ‘improve’ changed to ‘enhance’; ‘cold’ changed ‘freezing’; ‘sugar’ changed to ‘sugars’.
Introduction
Line 30: ‘environmental’ changed to ‘abiotic’. Delete ‘crop’.
Line 31: ‘elaborate’ changed to ‘complicated’; ‘to’ changed to ‘against’.
Line 33: ‘plant is a….main stages:’ changed to ‘plants need a dynamic development, including three development stages’.
Line 34: ‘following exposure’ changed to ‘after exposing’.
Line 35-36: ‘when plants …gradually lose’ changed to ‘plants will gradually lose freezing tolerance when placed under warm temperatures.’.
Line 37: ‘Seasonal changes in …’ changed to ‘Plants can undergo many physiological and biochemical variations underlying seasonal changes’
Line 39-40: ‘has been widely…’ changed to ‘studied in plants. Starch that is an important nutrient in plant growth and development is not only used for essential life activities, but also involves in responsing to abiotic stress.’ .
Line 41: ‘sugar’ changed to ‘sugars’.
Line 42-44: ‘extensively in many…’ changed to ‘Plants accumulate soluble sugars when they are exposed to cold temperatures, which can lower osmotic potential, maintain turgor pressure in dehydrated cells and allow plants to tolerate dehydrative stress’.
Line 44: ‘sugar’ changed to ‘sugars’.
Line 44-45: ‘protect the plasma membranes…’ changed to ‘safeguard function of the plasma membranes and proteins’.
Line 45: ‘…CA in Arabidopsis’ changed to ‘Arabidopsis plants showed sucrose accumulation during CA, a positive relation’
Line 47: ‘tolerance have’ changed to ‘tolerance had’; ‘sugar content’ changed to ‘sugars contents’.
Line 48: ‘sugar’ changed ‘sugars’; ‘β-amylase’ changed to ‘The β-amylase’.
Line 49-50: ‘Moreover, β-amylase ...9 BAMs’ changed to ‘Moreover, β-amylase has a typical sugar-binding 14 domain (PF01373) at the N-terminus, and is a member of glycosyl hydrolase family 14. Up till now, gene family analyses have identified 9 BAMs’.
Line 52: ‘gene’ changed to ‘genes’; ‘limited’ changed to ‘scarce’.
Line 53: ‘β-amylase’ changed to ‘BAM genes’;.
Line 54: ‘and was correlated with freezing tolerance‘ changed to ‘and activity of β-amylase had positive correlation with freezing tolerance’.
Line 55- 59: Add kiwifruit background.
Line 60: ‘comprises’ changed to ‘comprises on’.
Line 61: ‘A. arguta,…’ changed to ‘A. arguta which is‘; ‘down’ changed to ‘up’.
Line 63: ‘in’ changed to ‘on’.
Line 64: ‘could’ changed to ‘can’.
Line 65: ‘The knowledge on response of the Actinidia to cold stress’ changed to ‘The knowledge about response of cold stress on the Actinidia’.
Line 66: ‘the cold acclimation mechanism of’ changed to ‘mechanism of CA in’.
Line 67-71: whole paragraph changed to ‘In this study, we investigated carbohydrate metabolism and freezing tolerance under CA and DA stages of two cultivars for a better understanding of mechanism of freezing tolerance in kiwifruits.. Additionally, we analyzed AcBAMs genes family and relative expression of candidate gene to study mechanism of freezing tolerance in depth. We hope this study will provide theoretical basis for increasing freezing tolerance of kiwifruit.’.
Results
Line 74: ‘temperature’ changed to ‘temperatures’; ‘temperature data…’ changed to ‘Temperature data for China came from internet source’.
Line 75 : ‘…(www.weather.com.cn)...’ changed to ‘(www.weather.com.cn). Temperature of sampling day was calculated by taking daily mean’.
Line 76: ‘23th October’ changed to ‘23rd of October’.
Line 77: ‘22th’ changed to ‘22nd of’.
Line 79: ‘shape’ changed ‘shaped’.
Line 80: ‘changes’ changed to ‘variations’.
Line 81-84: put Table 1 into this position.
Line 86-92: the whole paragraph made a big modify.
Line 104: ‘meteorological’ changed to ‘air temperature’; ‘experimental’ changed to ‘sampling’.
Line 105: ‘cold hardiness’ changed to ‘freezing tolerance’.
Line 108-123: three paragraphs made a big modify.
Line 137: ‘sugar’ changed to ‘sugars’; ‘E’ changed to ‘F’.
Line 140: ‘overall’ changed to ‘all’.
Line 156: ‘is’ changed to ‘was’.
Line 175: ‘from’ changed to ‘by’.
Line 209: ‘belong’ changed to ‘belonged’; add ‘one’ before PbrBAM3, PtrBAM1, VcBAM3 and CsBAM3.
Line 227-228: genus name changed to italic type.
Line 232: ‘because of’ changed to ‘due to’.
Line 235: ‘1664’ changed to ‘1644’.
Line 264-270: the whole paragraph has mad a big modify.
Line 280: ‘ACTIN’ changed to ‘β-ACTIN’.
Line 282-283: moved table 3 into this position.
Discussion
Line 285-351: English grammar have mad a huge modify. The details were shown in revised manuscript.
Line 374: ‘filed’ changed to ‘yield’; ‘sugar’ changed to ‘sugars’.
Materials and Methods
Line 378: ‘comes’ changed to ‘came’.
Line 379: ‘is’ changed to ‘was’.
Line 380: ‘both different; changed to ‘each’.
Line 381: ‘Zhengzhou fruit… repository’ changed to ‘Zhengzhou Fruit Research Institute Kiwifruit Germplasm Repository’.
Line 392: add ‘approximate’.
Line 395: ‘boil’ changed to ‘boiling’; ‘cool’ changed ‘cooled’.
Line 406: add ‘approximate’.
Line 412-415: this paragraph has made big modify.
Line 417: ‘recognized’ changed to ‘identified.
Line 419: ‘question’ changed to ‘query’.
Line 421: add ‘was’.
Line 433: title changed to ‘Isolation and expression of genes’.
Line 444: added ‘subjected to’.
Line 445: added ‘software’.
Conclusions
Line 447: added ‘the’; ‘sugar’ changed to ‘sugars’.
Line 448: ‘driven’ changed to ‘attributed’.
Line 454: added ‘we suggested’.
The remianing corrections were shown in revised manuscript.
Line 458: added ‘and Abid Muhammad’.
Line 460: added ‘National Key Research and Development Project of China [2019YFD1000800];’.
We tried our best to improve the manuscript and made some changes in the manuscript. We appreciate your warm earnestly, and hope that the correction will meet with approval.
Once again, thank you very much for your comments and suggestions